# Developments on Core Collections of Plant Genetic Resources: Do We Know Enough?

Rui Gu , Shaohui Fan, Songpo Wei, Jiarui Li, Shihui Zheng and Guanglu Liu *

International Center for Bamboo and Rattan, Beijing 100102, China
* Correspondence: liuguanglu@icbr.ac.cn

**Abstract:** The core collection is a small subset that minimizes genetic redundancy while preserving the maximum genetic diversity of the entire population. Research on the core collection is crucial for the efficient management and utilization of germplasm resources. In this paper, the concept of the core collection and the research progress of its construction have briefly been summarized. Subsequently, some perspectives have been proposed in this research field for the near future. Four novel opinions have been presented, (1) the effective integration of multiple data types and accurate phenotyping methods need to be focused on; (2) the sampling strategy and bioinformatics software should be given attention; (3) the core collection of afforestation tree and bamboo species, with a wide natural distribution range and a large planting area, need to be carried out as soon as possible; (4) we should place a high priority on the study of genes discoveries and utilize these with a rapid, precise and high-throughput pattern based on re-sequencing technology. This paper provides a theoretical and technological reference for further study and the application of the plant core collection.

**Keywords:** germplasm resources; core collection; genetic diversity; utilization of core collection; molecular breeding



## 1. What Is a Core Collection?

Germplasm resources serve as a crucial material basis for genetic research and help in the identification and utilization of genes and traits that are of economic and ecological importance [1]. Therefore, the preservation and utilization of germplasm resources are of significant importance for the development of new crop varieties, and a large number of germplasm banks have been established [2]. However, due to their vast amount, diverse structure and incomplete information on germplasm resources, the available diversity that has been collected may not be fully and effectively utilized [3–5].

To obtain a germplasm bank that is both practical and representative, Australian scholars Brown [5] and Frankel [6] proposed the concept of core collection in 1984. Core collection refers to selecting a part of the entire germplasm resource through certain methods with the goal of representing the genetic diversity of the entire germplasm resource with a minimum number of resources. The theoretical basis supporting this concept is the theory of neutral mutations and the hierarchical structure model of genetic diversity [7]. A good core collection should have the following characteristics: representativeness, low redundancy, manageability, data completeness, and usability [8]. A core collection provides more reliable data and samples, makes it easier to optimize genotype/molecular marker-phenotype association studies, improves the utilization efficiency of the germplasm, and accelerates the breeding process [9–11]. Recently, with the continuous improvement in research methods, the concept of the core collection has been extended to the fields of asexual propagation crops, DNA germplasm banks and the in situ conservation research of germplasm resources [12].

## 2. The Progress of Core Collection

In considering world experience and the formation of core collections that the literature reveals, we focus on the following questions:

- Have core collections been formed for a diversity of plants?
- How can we effectively construct a representative core collection?
- How well can the core collection be utilized?

Our responses to these questions are summarized below.

### 2.1. Diversity of Core Collections

Core collections preserve the genetic diversity of the original population as much as possible, which promotes the effective use and protection of germplasm resources [4–6]. Based on this, many core collection research studies have been conducted both domestically and internationally. This study summarizes the development and research of core collections of 146 plant species over the most recent 10 years, which are listed in Table 1 below and Supplementary Table S1. The table shows that core collections have been developed mainly in economic crops and fruit trees; meanwhile, forages have been recently exploited for core collection establishment, including *Buchloe dactyloides* (Nutt.) [13], *Cynodon* Rich. [14] and *Bromus inermis* Leyss. [15]. However, core collections of endemic afforestation tree species are still limited, although some have been reported, such as those of *Cunninghamia lanceolata* (Lamb.) Hook. [16], *Robinia pseudoacacia* L. [17], *Populus tomentosa* Carrière. [18], *Pinus massoniana* Lamb. [19], etc. In addition, only a few of these studies have focused on spice crops, and the core collection that has been constructed is dominated by *Santalum album* L. [20].

**Table 1.** List of plant species that have been core collection-developed in recent years.

| Species Category | | Name |
|---|---|---|
| **Grain crops** | Cereals | maize [21–24], sorghum [25], coix [26], hulless barley [27], rice [28], wheat [9,29], oat [30], buckwheat [31], pearl millet [32], foxtail millet [33], peanut [34] |
| | Potatos | sweet potato [35], cassava [36] |
| | Pulses | chickpea [37], Pigeonpea [38], lima bean [39], soybean [40,41], rice bean [42], commom bean [43], faba bean [44], mung bean [45] |
| **Horticultural crops** | Vegetables | cauliflower [46], rapeseed [47], Cabbage [48], tomato [49,50], spinach [51], amaranth [52], bitter gourd [53], Jerusalem artichoke [54], yam [55], cucumber [56], pumpkin [57], white gourd [58], pepper [59], sweet pepper [60], eggplant [61], radish [62], Turnip [63], oyster mushroom [64], perilla [65], *Pyropia haitanensis* [66] |
| | Fruits | pricot [67–70], pear [71,72], jujube [73–75], grape [76], melon [77], watermelon [78], kiwifruit [79], pomegranate [80], litchi [81,82], olive [83], apple [84–86], peach [87], cherimoya [88], fig [89], sweet cherry [90], pomelo [91], persimmon [92], sugarcane [93] |
| | Ornamental plants | *Cymbidium ensifolium* [94], *Chrysanthemum morifolium* [95,96], *Prunus mume* [97], *Chimonanthus praecox* [98], *Rosa rugosa* [99], *Lilium brownii* [100], *Paeonia suffruticosa* [101], *Lagerstroemia indica* [102], *Helianthus annuus* [103], *Sophora moorcroftiana* [104] |
| | Herbs | *Fallopia multiflora* [105], *Astragalus* [106], *Scutellaria baicalensis* [107], *Angelica biserrata* [108], *Glycyrrhiza* [1], *Cornus officinalis* [109], *Dalbergia Odorifera* [110] |
| | Spice | *Santalum album* [20] |
| | Teas | Guizhou tea [111], Chinese tea [112,113] |
| **Beverages** | | Coffee [114], *Theobroma cacao* [115] |
| **Fibers** | | cotton [116], upland cotton [117], island cotton [118], ramie [119] |
| **Oilseeds** | | safflower [120], sesame [121] |
| **Forages** | | *Buchloe dactyloides* [13], *Cynodon* [14], *Medicago truncatula* [122], *Bromus inermis* [15] |
| **Trees** | | *Catalpa bungei* [123], *Catalpa fargesii* [124], *Saccharum spontaneum* [125], *Populus deltoides* [126], *Populus tomentosa* [18], *Cinnamomum camphora* [127], *Phoebe bournei* [128], *Robinia pseudoacacia* [17], *Torreya grandis* [129], *Tetracentron sinense* [130], *Xanthoceras sorbifolia* [131], *schima superba* [132], *Sapium sebiferum* [133], *Fraxinus chinensis* [134], *Eucommia ulmoides* [135], *Saccharum arundinaceum* [136], *Corylus avellana* [137], *Juglans regia* [138,139], *Betula platyphylla* [140], *Betula luminifera* [141], *Sinojackia huangmeiensis* [142], *Castanopsis hystrix* [143], *Morus alba* [144], *Castanea mollissima* [145], *Castanea sativa* [146], *Cunninghamia lanceolata* [16], *Cryptomeria japonica* [147], *Eucalyptus cloeziana* [148], *Eucalyptus urophylla* [149], *Ceratonia siliqua* [150], *Argania spinosa* [151], *Pinus massoniana* [19], *Pinus yunnanensis* [152], *Ginkgo biloba* [153], *Akebia trifoliata* [154], *Camellia oleifera* [155], *Cornus wilsoniana* [156] |

### 2.2. Procedure of Constructing a Core Collection

The development of the core collection has been extensively studied from various perspectives, such as sampling strategies, core size determination, and analysis methods, among others. However, due to the wide variation in the growth habits and reproductive characteristics of various plants, there is no universal core collection construction method. Generally, the construction of the core collection mainly includes four steps: the collection and organization of data, the grouping of accessions, the determination of sampling strategies and the testing and evaluation of the core set [1,8].

#### 2.2.1. Data Selection for Core Collection

Initially, passport data, phenotypic characteristics (agronomic and morphological traits), and biochemical traits were used to develop core collections, as these traits can visually represent plant differences and are straightforward to measure [37,157,158]. Based on these, core collections have been reported for many crops (Supplementary Table S1), including *Spinacia oleracea* L. [51] and *Chrysanthemum morifolium* Ramat. [95], *Helianthus annuus* L. [103], *Sorghum bicolor*(L.) Moench [25], *Solanum lycopersicum* L. [49] and *Coix lacryma-jobi* L. [26]. Additionally, this core collection method has been applied to many fruit trees, such as *Armeniaca vulgaris* Lam. [67], *Punica granatum* L. [80], *Ziziphus mauritiana* Lam. [75], *Prunus persica* L. Bat sch. [87], etc. Nevertheless, using phenotypic data produces certain challenges, such as its potential loss, incompleteness, unreliability, and easy susceptibility to environmental factors [17,124]. As a result, the core collection may not accurately represent the full genetic diversity of the original group, potentially resulting in the exclusion of important and valuable germplasms in the core collection [17,159].

With the increased availability and efficacy of molecular markers in uncovering genetic diversity, the development of more robust core collections has been made possible [124,139]. These markers include SRAPs for *Fallopia multiflora* (Thunb.) Nakai [105], *Sapium sebiferum* (Linn.) Roxb. [133]; ISSRs for *Chimonanthus praecox* (L.) Link [98], *Tetracentron sinense* Oliv. [130], *Cornus officinalis* Sieb. et Zucc. [109], *Argania spinosa* (L.) Skeels [151]; AFLPs for *Ginkgo biloba* L. [153], *Prunus mume* Sieb. et Zucc. J [97]; SSRs for *Raphanus sativus* L. [62], *Cunninghamia lanceolata* (Lamb.) Hook [16], *Actinidia chinensis* Planch. [79], *Astragalus membranaceus* Moench [106], *Olea europaea* L. [83], *Castanea mollissima* Bl. [149], *Corylus avellana* L. [137], etc. Additionally, SNPs for *Amaranthus tricolor* L. [52], *Brassica napus* L. [47], and *Litchi chinensis* Sonn. [81], *Glycine max* (L.) Merr. [41] and *Cucumis sativus* L. [56]. Among these various molecular markers, SSR markers are globally used, which can be attributed to their numerous benefits, including simple operation, high levels of polymorphism, a co-dominant inheritance, stability and reproducibility [160]. In recent years, SNP markers have become increasingly important in molecular detection, largely due to their high abundance in genomes and high-throughput genotyping on automated platforms [62].

Although molecular markers have several advantages in developing core collections, they cannot fully embody the genetic diversity of species due to their partial polymorphism of DNA fragments. Thus, breeders often employ multiple data types to construct a more robust and reliable core collection, which can greatly prevent the loss of crucial germplasms and enhance the precision and comprehensiveness of core sets [17,124]. For instance, Peng et al. [117] used phenotype, genotype and favorable alleles to develop a core collection of *Gossypium hirsutum* L. The core set of *Carthamus tinctorius* L. [120] and *Polygonum fagopyrum* L. [31] were established by molecular, phenotypic, and geographical diversity; Guo et al. [17] developed the *Robinia pseudoacacia* L. core collection using a combination of phenotype, physiology, and genotyping markers. The core collection of *Perilla frutescens* L. [65], *Catalpa fargesii f. duclouxii* (Dode) Gilmour [124], *Juglans regia* L. [139], *Castanopsis hystrix* Hook. f. and Thomson ex A. DC. [143], *Pinus yunnanensis* Franch. [152], *Camellia oleifera* Abel [155], etc., were developed by utilizing a combination of molecular markers and various phenotypic data.

### 2.2.2. Separating the Accessions into Meaningful Groups

To develop core collections effectively, it is crucial to create distinct and internally consistent groups that ensure representative sampling and reflect differences in genetic diversity. This can be achieved based on various factors such as plant taxonomy, geographical origin, ecological zone, phenotypic characteristics, and genetic distance [3,88]. Cluster analysis is a widely used tool for grouping accessions to construct core collections [161,162]. However, the choice of cluster methods greatly influences the resultant core collection, and several common cluster methods are available, such as single linkage, the complete method, the median method, the centroid method, and UPGMA and Ward's method [161]. Generally, the UPGMA method is widely used for its moderate nature and monotonicity [163]. Ward's method is also effective but requires the use of Euclidean distances. In addition to these commonly used methods, STAT software (StataCorp, College Station, TX, USA) provides other clustering methods, such as the maximum likelihood hierarchical clustering method (EML), density estimation method (DEN), and two-stage objective estimation method (TWO). Of course, the selection of a clustering method must be in combination with the corresponding sampling methods to improve the accuracy and rationality of the constructed core collection [163,164].

### 2.2.3. Sampling Strategies of Core Collection

The construction of core collection requires the careful consideration of sampling strategies to ensure maximal diversity and the minimal redundancy of selected accessions [4,16]. Sampling methods can be roughly divided into simple random sampling and stratification cluster sampling [4,8]: the former treats all germplasm materials equally, with a random selection of core accessions, ignoring the uneven distribution of genetic diversity and the frequency of different alleles in the entire genetic resource. Hence, this method may overlook certain accessions with low frequency but high variability in the entire group. By contrast, the stratification cluster sampling method, which is based on grouping and preserves the spatial and genetic distribution structure of original populations, is the most commonly used and effective method for the construction of core collections. It involves several steps to create a core collection. The first step is to determine the desired size of the core collection. Then, accessions can be grouped or stratified based on their characteristics, such as passport data, phenotypic data and genetic distance. Next various sample allocation methods, such as proportional (P), logarithmic (L), constant (C), square root (S), or genetic diversity-dependent allocation (G), are applied to determine the number of entries that are to be chosen from each group. Finally, one can either randomly select entries within each group or use a stepwise clustering approach to make the final selection [28,82,161]. Hu et al. [159] proposed a stepwise clustering method for selecting the final entries, where two materials were chosen at the lowest level of the clustering tree using random sampling, preferred sampling or deviation sampling. If there is only one germplasm, it is automatically included in the core set. The remaining materials were screened using the same method until the desired sampling size was achieved. Each of the three sampling strategies has unique characteristics [124,159]. Huang et al. [14] reported that the preferred sampling method was the most effective method for developing a core collection of *Cynodon* Rich, with both preferred and deviation sampling appearing more effective than random sampling in reducing the variance and coefficient of variation in the core collection [165]. Meanwhile, it is worth noting that cluster methods significantly affect the representation of core collections developed by three stepwise cluster sampling strategies [161]. Later, Wang et al. [161] proposed a minimum distance stepwise clustering strategy (LDSS) for selecting the final materials. This method selected one of two materials with the smallest genetic distance on the clustering diagram each time, resulting in a more effective reduction in genetic redundancy compared to other methods [161]. Figure 1 shows a diagram of the methodology used to establish the core collection.

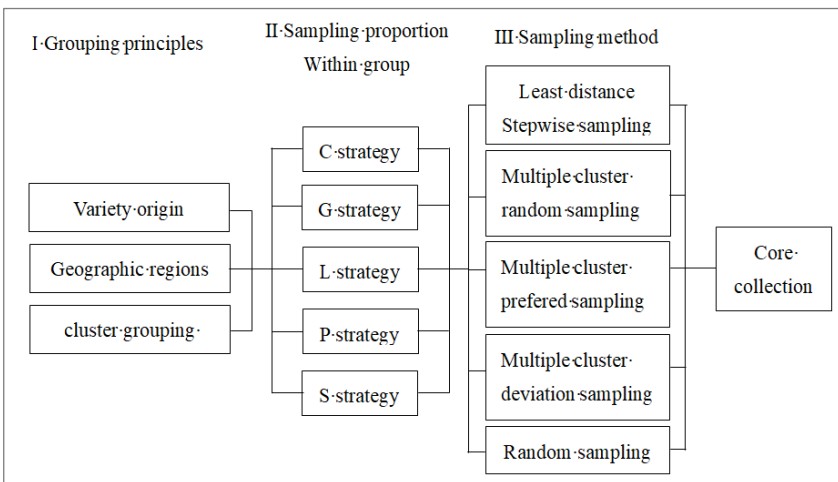

**Figure 1.** Stratification cluster sampling scheme of core collection for plant species.

In recent years, the maximization (M) strategy, which selects materials with a high allele abundance and low redundancy by maximizing the number of alleles at each locus, has been widely applied in core collection construction [16,166]. This approach does not rely on grouping or clustering methods, and the sampling proportion is automatically generated [166,167]. Currently, there are several software programs available for the construction of core collection based on the M strategy [16,137]: Power Mstrat [168], Power Core [169], CoreFinder [170] and Core Hunter [171,172]. Some studies have found that with the same sampling ratio, PowerCore retained a higher percentage of alleles and genetic diversity compared to other software programs [1,119]. However, Chen et al. [173] compared the PowerCore and the CoreFinder algorithms and showed that the core collection developed by CoreFinder was more suitable for NARO rapeseed (*Brassica napus* L.). While the M strategy often selects poorly representative germplasm due to the need to maximize allelic diversity, the distance-based method selects materials with good representativeness but may not be ideal in terms of allelic retention. To address this issue, Core Hunter software (Department of Computer Science, University of British Columbia, Vancouver, Canada) optimized genetic distance and allelic diversity simultaneously by weighting the modified Rogers distance and Shannon diversity index differently based on two optimization criteria [172].

Figure 2 and Supplementary Table S1 provide an overview of the sampling strategies used to develop core collections, which show that multiple cluster sampling based on a genetic structure is the most widely used, followed by the M strategy. I think that the optimal core collection should exhibit two characteristics: maximum genetic distance and maximum genetic diversity. The M strategy focuses on selecting diverse loci, while the genetic distance method aims to select diverse germplasm. Combining both approaches may lead to the best results. Of course, different plants are suitable for different sampling methods, and the strategy to be adopted can usually be determined by the purpose of the core collection [16,137].

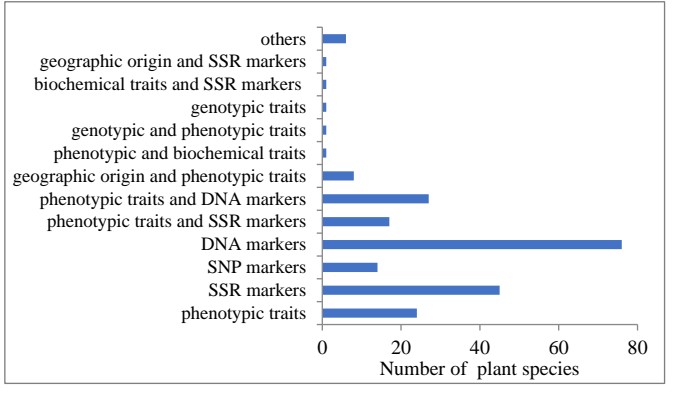
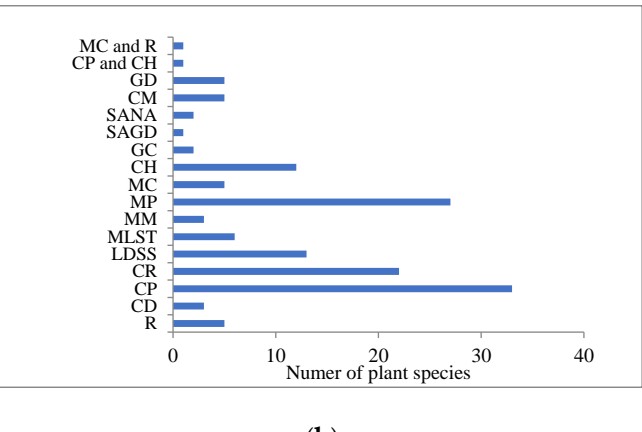

(**a**)                                                (**b**)

**Figure 2.** Statistical results of data collection and sampling strategies for core collection construction of 146 plant species. (**a**) Data collection; (**b**) Sampling strategies. Here, *R* random sampling, *CD* Multiple cluster deviation sampling, *CP* Multiple clusters preferred sampling, *CR* Multiple cluster random sampling, *LDSS* Least distance stepwise sampling, *MLST* Maximum length sub-tree method, *MM* M strategy algorithm of MSTRAT, *MP* M strategy algorithm of PowerCore(National Institute of Agricultural Biotechnology, Suwon, South Korea), *MC* M strategy algorithm of CoreFinder, *CH* Core Hunter method, *GC* GenoCore method, *SAGD* Simulated annealing algorithm based on the maximizing genetic diversity, *SANA* Simulated annealing algorithm based on maximizing allelic richness, *CM* Multiple cluster most representative sampling, *GD* Genetic diversity based on Genetic Subsetter, *CP and CH* Multiple clusters preferred sampling combined with Core Hunter method, and *MC and R* M strategy algorithm of Core Finder combined with Random sampling method are represented.

### 2.2.4. Sampling Proportion of Core Collection

A suitable sampling ratio is another critical factor to consider in the process of constructing core collections. A high sampling ratio may result in the inclusion of redundant samples in the core collection, while a low sampling ratio may lead to the loss of key materials [17]. Brown et al. [4] proposed that a sampling ratio of 10% could represent 70% of the genetic diversity in the original collection when the number of original collection samples was no less than 3000. Van Hintum et al. [3] argued that sampling proportions should range from 5 to 20%. Yonezawa et al. [174] suggested that a sample proportion of 20%–30% would be appropriate. In addition, germplasm with excellent traits, as well as backbone parents or varieties that have played a significant role in production, should be directly selected into the core collection to avoid the loss of excellent germplasm or genes.

However, Supplementary Table S1 shows that the sampling proportion for forming a core collection is generally about 1.70%–66.46% of the original collection, and the size of core collections can vary significantly, with some consisting of as few as 12 samples while others include up to 1956 samples. Clearly, the sampling ratio is not fixed but varies depending on the characteristics of the plant. The optimal fraction depends largely upon the original germplasm size, germplasm accessibility, germplasm similarity, and sampling strategy [7]. Under any circumstances, the preservation of a significant proportion of germplasm diversity should be the main consideration when determining the optimal fraction [148].

### 2.3. Evaluation of Core Collection

While the core collection is constructed based on available data, the important question remains: does the core set accurately represent the diversity of the original population?

Brown proposed that the core collection should represent 70% or more of the trait characteristics and genetic variations of the entire germplasm [4]. To validate the effectiveness of the core collection, it should be evaluated from two aspects: firstly, to test

the representativeness of the genetic diversity of the entire collection and, secondly, to assess its practicality in production [59]. Generally, at the molecular level, the main genetic diversity indices include the allele number (Na), effective allele number (Ne), Shannon's information index (I), Nei's genetic diversity index (H), polymorphism information content (PIC), observed heterozygosity (Ho) and expected heterozygosity (He) [159]; among these, allelic richness is considered the most relevant indicator. Maximizing allelic richness means preserving the germplasm resource with the most abundant genetic diversity. At the phenotypic level, the evaluation parameters include the mean difference percentage (MD), variance difference percentage (VD), coincidence rate of range (CR) and variation coefficient changing rate (VR) [8,69,159]. Usually, the core collection is considered representative only when the MD is less than 20%; the CR is more than 80% [159]. A lower value in MD and a higher value in VD, CR, and VR could be considered to indicate a more representative core collection [14,59]. In addition, principal component analysis (PCA) plots have been widely used to compare the distribution characteristics between the core collection and the initial population [14]. Moreover, correlation analysis is commonly conducted to infer whether the inherent relationship between traits in the original collection is well retained in the core group [14]. Recently, Odong et al. [8] proposed two new criteria based on genetic distance to evaluate the quality of the core collections. These criteria offer the advantage of simultaneously considering all variables describing the accessions and provide intuitive and interpretable results compared to the univariate criteria generally used in core collection evaluations. Additionally, after establishing a core collection, it is essential to establish a comprehensive management system for breeding, seed supply, and exchange as soon as possible to ensure the distribution, sharing, and effective utilization of the core set.

In short, the evaluation criteria of core collection should be variable, and flexible evaluation methods should be tried according to the new situation. The selection of the most suitable evaluation method should depend upon the purpose of core collections [8]. Moreover, core collection establishment is a dynamic process [148] that needs to be regularly updated by the addition of new entries and the removal of duplicates to improve representativeness and maintain dynamism [120].

## 3. The Use of Core Collection

The main purpose of constructing core collections is to effectively address the conflict between a large number of germplasm resources in gene banks and their effective preservation, to improve the utilization rate of excellent germplasm, to promote the exploration of excellent gene and molecular markers, and to achieve innovation and genetic improvement in crop germplasm [17,175]. Currently, with the development of next-generation sequencing technology, domestic and foreign research institutions are conducting deep resequencing of the core collection to accelerate breeding efficiency.

### 3.1. Genomic Study and Marker Development

Molecular markers are valuable tools in studies such as genotype determination, genetic diversity analysis, marker-assisted breeding, the fine mapping of QTLs, and phylogenetic analysis. Due to the existence of reference genome sequences for various species and their access to core collections, it is now possible to create a vast number of DNA markers, such as SNP, insertions/deletion (InDel), cleaved amplified polymorphic sequence (CAPS), etc. Molecular markers have broad prospects for application in the core collection, which can promote the protection and utilization of germplasm resources.

#### 3.1.1. Evaluation of Genetic Diversity

Developing molecular markers for core collection is beneficial for evaluating genetic variation, gaining insights into the genetic background of germplasm resources, and thus protecting and enhancing genetic diversity. For example, Wu et al. [176] used genotyping-by-sequencing to identify 4680 high-quality SNPs in a core collection of 150 Chinese *Ziziphus jujuba* Mill. accessions, and characterized their genetic diversity, population structure, and

linkage disequilibrium based on SNPs. Additionally, Lu et al. [177] developed 23 novel markers using restriction site-associated DNA sequencing to analyze the genetic diversity of *Amomum tsao-ko*.

### 3.1.2. Classification of Species and Varieties

Molecular markers can help identify the species and varieties of core collections that are difficult to distinguish morphologically. Li et al. [178] designed 134 primers based on cucumber genome re-sequencing, with 116 showing polymorphisms in 16 germplasms, which were used for the identification and evaluation of cucumber. Similarly, Guo et al. [179] identified InDels throughout the genome of *Capsicum* spp. and developed three PCR-based markers that could distinguish interspecific hybrids of five domesticated *Capsicum* species (*C. annuum*, *C. chinense* Jacq., *C. baccatum*, *C. pubescens* Ruiz and Pavon and *C. frutescens*).

### 3.1.3. Construction of Genetic Maps

Molecular marker analysis can construct genetic maps of germplasm resources, which help understand their genetic structure and evolutionary history, providing a basis for comparative genomics research. For example, Meng et al. [180] aligned the re-sequencing results of two *Brassica campestris* accessions, 'R-O-18' and 'L58', to the genome sequence of 'Chiifu-401-42', and developed 99 InDel markers to construct a genetic linkage map of *Brassica rapa*. Yu et al. [181] genotyped 150 recombinant *Brassica rapa* inbred lines using genome resequencing and generated a high-density genetic map containing 2209 SNPs. They observed a high recombination rate of up to 20% for the genetic markers and detected 18 QTLs for the leafy head traits and three candidate genes. In addition, Song et al. [182] performed resequencing on six non-heading Chinese cabbage variants(Pak-choi, Taitsai, flowering Chinese cabbage, turnip, Tillering cabbage, and Wutatsai) and analyzed the distribution and quantity differences of SSRs and SNPs among these variants, which provided important marker information for the construction of high-density genetic maps and conducted comparative genomics among non-heading Chinese cabbage variants.

### 3.1.4. Discovery of Specific Genes

Currently, novel molecular markers derived from functional genes play a crucial role in improving agronomic characteristics such as yield, quality, disease resistance, and abiotic stress tolerance. For instance, Kim et al. [183] developed functional markers for Ty-2 and Ty-3 resistance in yellow leaf curl virus-resistant tomato varieties, with two InDels and three CAPS markers, which were developed for Ty-2 and Ty-3, respectively, to aid in the integration of Ty-2 and Ty-3 genes into high-performing tomato cultivars. Similarly, Qing et al. [184] developed the InDel marker "MM28T" for the brown planthopper (BPH) resistance gene (*Bph3*) in rice to produce BPH-resistance rice cultivars and reduce damage caused by BPH. Furthermore, Kroc et al. [185] developed the co-dominant-derived cleavage amplification polymorphic sequence (dCAPS) marker "iuc_RAP2-7", utilizing SNPs in RAP2-7: a candidate gene linked to the seed alkaloid content in narrow-leafed lupin (*Lupinus angustifolius* L.) to produce low-alkaloid cultivars. Additionally, Liu et al. [186] and Wang et al. [187] developed SSR and Indel markers using the whole-genome sequencing of 9930 and re-sequencing data of 100 or 115 core cucumber germplasms in the primary mapping region of the black spine color gene or heavy netting gene, respectively. Their study revealed that the black fruit spine trait and the heavy netting trait were controlled by one dominant nuclear gene, (B) and (H), respectively, that were located on chromosomes 4 and 5 in cucumber. These findings were crucial for the fine mapping and gene cloning of *B* and *H* genes in cucumbers.

Collectively, these novel molecular markers provide an effective approach for classifying germplasm resources, conducting genetic research, identifying the specific genes affecting important traits, and facilitating marker-assisted breeding.

### 3.2. Identification of Disease or Pest Resistance

Core collections, which represent the entire genetic diversity of germplasm resources, have been shown to improve the efficiency of identifying disease or pest-resistant accessions or genes. In recent years, many researchers have conducted studies on disease or pest resistance in crop core collections, and a list of some of the more common disease or pest-resistant QTLs/genes identified is provided in Table 2.

**Table 2.** Disease and the number of resistant QTLs/genes/resistant samples in representative crop core collections.

| Core Collection | No. of Sample | Type of Disease | No. of QTLs/Genes/ Resistant Samples | Reference |
|---|---|---|---|---|
| rice | 150 | sheath blight | 13 genes | [188] |
| rice | 510 | bacterial leaf streak | 69 QTLs | [189] |
| maize | 183 | fusarium ear rots | 4 genes | [190] |
| wheat | 121 | tan spot *Pt* race 1 | 10 QTLs | [191] |
| | | tan spot *Pt* race 5 | 5 QTLs | |
| | | stagonospora nodorum blotch | 5 QTLs | |
| wheat | 331 | leaf rust | 11 QTLs | [192] |
| | | powdery mildew | 58 resistant accessions | |
| | | leaf rust | 42 resistant accessions | |
| barley | 159 | net blotch | 2 resistant accessions | [193] |
| | | mild mosaic virus | 13 resistant accessions | |
| | | yellow dwarf virus | 32 resistant accessions | |
| sorghum | 318 | anthracnose | 4 QTLs | [194] |
| | | anthracnose | 13 resistant accessions | |
| sorghum | 242 | leaf blight | 27 resistant accessions | [195] |
| | | leaf rust | 6 resistant accessions | |
| | | anthracnose, leaf blight, leaf rust | 3 resistant accessions | |
| common bean | 211 | powdery mildew | 4 genes | [196] |
| common bean | 315 | cyst nematode HG Type 0 | 11 SNPs | [197] |
| common bean | 168 | bacterial wilt | 14 genes | [198] |
| mungbean | 296 | dry root rot | 29 resistant accessions | [199] |
| cowpea | 375 | *Aphis craccivora* Koch pest | 3 resistant accessions | [200] |
| pigeonpea | 146 | fusarium wilt | 6 resistant accessions | [201] |
| | | sterility mosaic | 24 resistant accessions | |
| | | fusarium wilt and sterility mosaic | 5 resistant accessions | |
| lentils | 188 | pea aphid | 14 genes and 13 SNPs | [202] |
| groundnut | 213 | groundnut rosette | 32 SNPs | [203] |
| peanut | 99 | aflatoxin | 16 SNPs | [204] |
| upland cotton | 419 | verticillium wilt | 12 resistant accessions | [205] |
| flax | 447 | powdery mildew | 1 gene and 3 QTLs | [206] |
| oilseed rape | 166 | blackleg | 8 QTLs | [207] |
| tomato | 171 | yellow leaf curl virus | 2 genes | [183] |
| apple | 176 | apple scab | 10 genes | [208] |
| watermelon | 35 | fusarium wilt | 1 SRAPs | [209] |
| melon | 4 | powdery mildew | 112 SNPs and 12 InDels | [210] |
| safflower | 84 | fusarium wilt | 3 AFLPs and 1 SSRs | [211] |

Mini-core collections of various crops have been evaluated to identify sources of resistance against different diseases or pests. For instance, in the mungbean, three accessions (VI001509AG, VI001400AG, and VI001244AG) were found to be resistant to most isolates of dry root rot out of 296 evaluated accessions [199]. In pigeonpea, six, twenty-four, and five accessions were found to be resistant to fusarium wilt and sterility mosaic disease, with combined resistance, respectively, out of 146 evaluated accessions [203]. In the cowpea, three genotypes (TVu-6464, TVu-1583, and TVu-15445) were found to be resistant to *Aphis craccivora* Koch [200]. In watermelon, Yang et al. [209] analyzed the genetic variation and

relationships of 35 core collections and identified one SRAP locus that was associated with fusarium wilt resistance.

Furthermore, GWAS studies have also been conducted to identify novel sources of disease resistance by exploiting a natural variation in crop core collections. For example, Fu et al. [188] identified sheath blight disease resistance in Ting's rice core collection of 150 varieties and discovered 13 resistance gene-based SNPs from approximately five million SNPs. A core set of 121 bread wheat accessions were evaluated for their resistance against the tan spot and the stagonospora nodorum blotch (SNB); then, ten, five, and five genomic regions were associated with resistance to tan spot race 1, race 5, and SNB, respectively [191]. In the common bean core collection, Binagwa et al. [196] identified several SNPs which were responsible for powdery mildew disease resistance, revealing four putative resistance genes. Shi et al. [197] evaluated 315 accessions for soybean cyst nematode disease resistance and observed 11 SNPs in HG Type 0 responsive chromosomes. In the African core groundnut collection, 32 SNPs were associated with the detected groundnut rosette disease resistance [203], two of which were located on the exons of a putative TIR-NBS-LRR disease resistance gene.

In conclusion, core collections provide a valuable genetic resource for the identification of disease-resistant QTLs/genes, which can greatly aid in disease-resistance breeding efforts and may even lead to gene editing and the introduction of novel alleles for single or multiple disease resistance in various crops.

*3.3. Gene Discovery and Allele Mining*

Core collections are valuable genetic resources for the identification of elite genes and mining alleles. Table 3 provides information on the characteristics of marker-trait associations in the core collections of representative crops.

**Table 3.** Marker-trait association using core collections in representative crops.

| Core Collection | No. of Sample | Trait | No. of QTLs/Genes | Reference |
|---|---|---|---|---|
| rice | 150 | salt tolerance | 65 QTLs | [212] |
| rice | 191 | yield and heavy metal content | 250 QTLs | [213] |
| | | agronomic traits | 97 QTLs | |
| rice | 150 | agronomic | 32 QTLs | [214] |
| rice | 150 | cold tolerance | 26 QTLs | [215] |
| soybean | 224 | agronomic | 16 QTLs | [216] |
| soybean | 189 | yield and yield component | 19 QTLs | [217] |
| soybean | 23 | salt tolerance | 22 QTLs | [218] |
| | | low temperature tolerance | 15 QTLs | |
| | | high oil content | 6 QTLs | |
| soybean | 159 | high protein content | 1 QTLs | [219] |
| | | drought tolerance | 5 QTLs | |
| soybean | 46 | low temperature resistant | 13 QTLs | [220] |
| wheat | 568 | yield and yield components | 17 QTLs | [221] |
| upland cotton | 419 | fiber-related | 5 genes | [222] |
| leaf heads of cabbage | 150 | agronomic | 18 QTLs | [181] |
| tomato | 360 | agronomic | 2 QTLs | [223] |
| melon | 79 | fruit quality | 241 QTLs | [224] |
| sugarcane | 97 | yield components and sucrose | 56 QTLs | [225] |

Many genes that affect agronomic traits have been discovered in rice core collections [226–229]; these include *Rc*, *waxy*, and *GS3* genes, which were found to be associated with grain color, amylose content, and grain length, respectively [230]. In addition, 65 QTLs and four candidate genes (*LOC_Os06g47720, LOC_Os06g47820, LOC_Os06g47850, LOC_Os06g47970*) were identified for salt tolerance [212]. Soybean core collections have also been genotyped using SNP markers to study linkage disequilibrium and associations with agronomic, yield and yield components; 12 genes, 19 SNPs, and five haplotypes were

identified and linked to these traits [216,217]. QTLs in relation to low temperature, salt tolerance, and drought stress were also identified [218–220]. Additionally, genes that influenced upland cotton fiber quality, such as, *GhCIP1* and *GhUCE* for days to flowering, *GhFL1* and *GhFL2* for fiber length, and *Gh_A07G1769* for fiber strength, were discovered [222]. In vegetable and fruit core collections, various genes controlling key fruit traits have been identified. For example, Qi et al. [231] conducted deep resequencing on 115 cucumber accessions to generate a genetic map and revealed the domestication and diversity of cucumbers; they identified ~3.6 million variants, including a natural genetic variant in a β-carotene hydroxylase gene that could improve the nutritional value of cucumbers. Lin et al. [223] constructed a genetic map of tomato genome variation based on the genome sequences of 360 core accessions and discovered two independent QTLs (*fw2. 2* and *lcn2. 1*), which resulted in modern tomato fruits that were 100 times larger than their wild counterparts. They also identified a key causative variant site in the promoter region of the *SlMYB12* gene that conferred the pink fruit color. Moreover, studies on melons have identified a candidate white-flesh gene (*CmPPR1*) that affects fruit flesh color and a ThAT gene (*CmThAT1*) that mediates thioester production [224]. In Louisiana, sugarcane core collection analysis has discovered 56 markers to be consistent with cane yield components and sucrose traits [226]. Guo et al. [232] reported a high-quality draft genome sequence of the east Asia watermelon cultivar '97103' and resequenced 20 accessions of watermelon from three different subspecies(*lanatus*, *Mucosospermus* and *vulgaris*), revealing the loss of many disease resistance genes during domestication and identifying several genes that were related to valuable fruit-quality traits, including sugar accumulation and citrulline.

In summary, functional genes or markers can improve the efficiency and precision of selecting desirable traits and aid in accumulating favorable alleles for high-yield crop breeding. Further work is necessary to mine more underlying genes, manipulating the desired traits.

## 4. Conclusions and Perspectives

Considering the current status of core cultivation, the following aspects deserve attention.

### 4.1. Effective Data Integration and Accurate Phenotyping Methods

The use of molecular markers in conjunction with phenotypic and passport data are becoming increasingly common in the establishment of core sets, but how to scientifically and effectively integrate these multiple types of data is still a problem faced by the construction of core collection, which needs more attention. In addition, phenotyping is the key to core collection success construction, as phenotypes are regarded as the most reliable indicators for genotypes. Therefore, it is necessary to improve phenotyping tools to enhance the efficiency of phenotyping.

### 4.2. Appropriate Sampling Strategy and Software

Optimizing the sampling strategy is crucial for constructing a core collection that is both representative and effective. Moreover, the successful construction of the core collection has been accompanied by the development of different software Hence, on the basis of mastering multiple types of data, the type of sampling strategies and software to use has become a hot spot and focus of future core collection research.

### 4.3. Broaden the Scope of Core Collection

Currently, as the number of endangered tree species and low-quality tree species gradually increases, there is an urgent need to collect and establish a core collection for the protection and effective utilization of germplasm resources. However, core collections are mainly concentrated on grain crops or horticultural plants. Few studies have focused on afforestation and endangered tree species: this is especially true for bamboo species. The

development of core collections for timber trees and bamboo, especially those native and special species in China, is a top priority for the near future.

### 4.4. Identify Abundant Functional Genes and Markers

The core collection is essentially the gene bank of a species, and one of the most important goals in constructing core sets is to explore and utilize high-quality genes, such as those for high yield, good quality, resistance to disease and pests and stress tolerance. Currently, the widespread application of second-generation sequencing technologies has made it possible to re-sequence every resource in the core collection. Considering the wide variety and rich genetic diversity of plants that exist, in the future, we should make full use of genome sequencing information to develop large-sample, low-abundance genotyping methods and technologies and aim to improve the efficiency of molecular markers and gene discovery based on the core collection.

**Supplementary Materials:** The following supporting information can be downloaded at: https://www.mdpi.com/article/10.3390/f14050926/s1, Table S1: Core collection construction of partial plant species.

**Author Contributions:** Conceptualization, S.F. and S.W.; methodology, S.W.; formal analysis, S.Z.; data curation, J.L.; writing—original draft preparation, R.G.; writing—review and editing, R.G. and G.L.; visualization, G.L.; supervision, G.L.; project administration, G.L.; funding acquisition, G.L. All authors have read and agreed to the published version of the manuscript.

**Funding:** This study was supported financially by Important bamboo and rattan germplasm biomass formation geographical differentiation, the National Key Research and Development Project of China of the 14th Five-Year Plan (Grant Number 2021YFD2200501).

**Data Availability Statement:** The data are included in the article or Supplemental Materials.

**Conflicts of Interest:** The authors declare no conflict of interest.

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
