# Peer review of "Developments on Core Collections of Plant Genetic Resources: Do We Know Enough?"

_forests, doi:10.3390/f14050926_

Round 1
Reviewer 1 Report
The review paper of Gu et al. is well-presented and the English is clear and easy to read. The document is well structured and most of the different sections are appropriated. However, some sections need to be improved, adding more information. It is a review paper and you should give a more complete detail (specially in section 3.1; 3.2 and 3.3 - see comments).
Some modifications need to be taken into account as follows:
Section 2.2.3, Lines 136-158: You have mentioned that several methods to form core collection have been proposed including simple random sampling and stratification cluster: You should add a summary table mentioning all the different methods used and what are the differences between them, Pros and Inconvenient de each method.
Besides, you only explained the steps of stratification cluster, and what about the simple random sampling.
Figure 2:
In the ‘X’ axis of this Figure 2, you should add the legend of this axis (Sample % or what?)
Line 192: Correct “strategies” instead of “strateries”
Line 193: The Statistical results of this Figure, represents data of 146 plants? or 146 crop species!! Please specify
Line 206: ‘different plants’ it is better to use ‘crop species’ or ‘plant species’ depending
Line 268: the subtitle “Improving utilization efficiency” should be changed it does not represent what you detailed in this paragraph. Maybe you can use this subtitle ”Identification of disease resistance” or “gene resistance” or Exploring new genes” something like that. Besides, in this section I suggest to add a table with the different resistance identified in the different core collection and the different genes. It is a review paper and you should put more information.
Developing core collections representing the diversity and evaluating a common set of lines under different set of environmental conditions would enhance the utility of crop species germplasm.
Line 271: delete ‘significantly’
Line 276: the sentence is unclear, it is better delete ‘core accessions’
Line 282: ‘is’ instead of ‘was’
Line 297: ‘constructed a map’: you should specify if genetic or physical map
Line 298: I think that ‘genome variation’ should be removed
Line 299: ‘collected from around’ you should specify origin or just mention collected from where
Line 300: ‘larger than its ancestor’ : please add the detail of tomato ancestor
Line 311: ‘ C. lanatus’ should be in itallic. Besides you mentioned ‘three differeent C.lanatus subspecies’ please add the detail of these 3 subspecies.
Line 313: In the section 3.3 ‘Developing DNA markers’ : You should add more species and add a table representing the crop, the developed markers and the methodology used to give to readers a general and complete view of what was obtained.
Same coments for section 3.2 (more information needed or just mention the information in a summary table)
Line 321: delete ‘C.sativus’ as it is already mentioned in line before.
Line 337: I suggest to change the subtitle of this section! Something like “Challenges and perspectives” or something else. Besides, you should structure more this section (338 to 364) and developp it.
Line 345: you mentioned “….by the development of different softwares and bioinformatics tools”, be specific and cite the most used ones.
Supplementary table S1:
In the footnote of this table, you should add the explanation of the abreviations used in column of sampling method.
Reviewer 2 Report
1. Author should provide some flow charts in better understanding of his basic concepts about diversity, core collections etc.
2. Minor spelling and grammatical mistakes should be rectified with in the manuscript.
3. Write more about core collection through molecular approach (Representation through table for well understanding)
4. Any Bioinformatic/in-silico approach was not mention for any core collection and its applied works.
5. Author should give a table for number of plants accessions of core collection developed/identified till now.
Reviewer 3 Report
This well-prepared and informative review adequately discusses the theoretical and technological references for further study and application of plant core collection.
Basically, plant genetic resources are characterized to reveal genetic differences between seed samples or populations and the amount and distribution of genetic variation in these samples and populations. By means of genetic markers, which of the seed samples stored in duplicate in gene banks has priority for conservation, the optimum sizes of their populations, core collections, and gene flows are determined.
Molecular markers make it possible to analyze a whole genome with DNA obtained from a very small amount of tissue from the plant. They have been used extensively in germplasm characterization in recent years with these techniques. Thus, plant genetic resources have begun to be characterized more accurately and precisely. Here are some points I want you to focus on in this direction:
(1) Instead of considering these marker systems as alternatives to morphological markers, it would be a more accurate approach to consider them as complements. How can you explain this in more detail in future aspects (lines 337-364)?
(2) In my opinion, the authors should make an evaluation according to the IUCN categories in order for this study to be more sound. Are there any plants that you are trying to highlight in your review that rank critically according to IUCN categories? If so, could you integrate the importance of their IUCN categories and conservation suggestions into the article?
(3) Out of a total of 197 references, 82 (41%) belong to the last 5 years. If possible, could you increase this to a higher percentage (near 50%)?
(4) English language and style in some parts of the review require fine/minor spelling, so authors should check thoroughly to make these minor corrections.
